# Predictors of drug survival for biologic and targeted synthetic DMARDs in rheumatoid arthritis: Analysis from the TRA Clinical Electronic Registry

Ching-Tsai Lin[1], Wen-Nan Huang[1], Wen-Chan Tsai[2], Jun-Peng Chen[3], Wei-Ting Hung[1,4], Tsu-Yi Hsieh[1,4], Hsin-Hua Chen[1,3,5,6], Chia-Wei Hsieh[1,6], Kuo-Lung Lai[1], Kuo-Tung Tang[1,6], Chih-Wei Tseng[1], Der-Yuan Chen[7,8], Yi-Hsin Chen[1,5]*, Yi-Ming Chen[1,3,5,6]*

1 Division of Allergy, Immunology and Rheumatology, Taichung Veterans General Hospital, Taichung, Taiwan, 2 Department of Internal Medicine, Kaohsiung Medical University Hospital, Kaohsiung Medical University, Kaohsiung, Taiwan, 3 Department of Medical Research, Taichung Veterans General Hospital, Taichung, Taiwan, 4 Department of Medical Education, Taichung Veterans General Hospital, Taichung, Taiwan, 5 School of Medicine, College of Medicine, National Yang Ming Chiao Tung University, Taipei, Taiwan, 6 Rong Hsing Research Center for Translational Medicine & Ph.D. Program in Translational Medicine, National Chung Hsing University, Taichung, Taiwan, 7 Rheumatology and Immunology Center, China Medical University Hospital, Taichung, Taiwan, 8 School of Medicine, China Medical University, Taichung, Taiwan

* ymchen1@vghtc.gov.tw (YMC); ysanne@vghtc.gov.tw (YHC)

**Data Availability Statement:** All relevant data are within the manuscript and its Supporting Information files.

## Abstract

In this study we aimed to identify the predictors of drug survival for biologic and targeted synthetic DMARDs (bDMARDs and tsDMARDs) among patients with rheumatoid arthritis (RA) in a real-world setting. Data from RA patients receiving bDMARDs and tsDMARDs between 2007 and 2019 were extracted from the Taiwan Rheumatology Association Clinical Electronic Registry (TRACER). Patients were categorized into tumor necrosis factor-alpha (TNF-α) inhibitors, non-TNF-α inhibitors, and tofacitinib groups. The primary outcome was 3-year drug retention and the causes of bDMARDs and tsDMARDs discontinuation were recorded. Baseline demographic data before the initiation of bDMARDs and tsDMARDs treatment were analyzed to identify the predictors of 3-year drug survival. A total of 1,270 RA patients were recruited (TNF-α inhibitors: 584; non-TNF-α inhibitors: 535; tofacitinib: 151). The independent protective factors for 3-year drug survival were positive rheumatoid factor (RF) (HR: 0.48, 95% CI: 0.27–0.85, $p = 0.013$) and biologics-naïve RA (HR: 0.61, 95% CI: 0.39–0.94, $p = 0.024$). In contrast, positive anti-citrullinated protein antibody (ACPA) (HR: 2.24, 95% CI: 1.32–3.79, $p = 0.003$) and pre-existing latent tuberculosis (HR: 2.90, 95% CI: 2.06–4.09, p<0.001) were associated with drug discontinuation. RA patients treated with TNF-α inhibitors exhibited better drug retention, especially in the biologics-naïve subgroup ($p = 0.037$). TNF-α inhibitors were associated with lower cumulative incidence of discontinuation due to inefficacy and adverse events (both $p<0.001$). Baseline RF and ACPA positivity in abatacept-treated patients were associated with a better 3-year drug survival. However, negative ACPA levels predicted superior drug survival of TNF-α

**Funding:** This study was supported by a grant from Taichung Veterans General Hospital, Taiwan (TCVGH-1087312C) to Dr. Yi-Ming Chen. The funder has no role in the study design, data collection and analysis, decision to publish, or preparation of the manuscript.

**Competing interests:** The authors have declared that no competing interests exist.

inhibitors and tofacitinib. In conclusion, bio-naïve status predicted better drug survival in TNF-α inhibitors-treated RA patients. RF and ACPA positivity predicted better abatacept drug survival. In contrast, ACPA negativity was associated with superior TNF-α inhibitors and tofacitinib survival.

## Introduction

Rheumatoid arthritis (RA) is a chronic and debilitating form of arthritis, and is one of the most prevalent autoimmune inflammatory rheumatic diseases [1]. According to the American College of Rheumatology (ACR)/European League Against Rheumatism (EULAR) recommendations for management of RA, the aim of treatment should be to reach a target of sustained remission or low disease activity in every patient with either biologic disease-modifying anti-rheumatic drugs (bDMARDs) or targeted synthetic DMARDs (tsDMARDs) [2, 3]. Advances in targeted therapy of RA have shown efficacy in preventing bone erosion and joint deformities [4]. However, the optimal response rates among bDMARDs and tsDMARDs for achieving low disease activity and remission were shown to be no greater than 50% and 20%, respectively [4, 5]. Therefore, it is imperative to identify predictors of drug retention for bDMARDs and tsDMARDs in RA using registries and health care databases [6–10].

The Taiwan Rheumatology Association Clinical Electronic Registry (TRACER) is a prospective, non-randomized cohort that promotes the "treat-to-target (T2T)" strategy for RA nationwide. In Taiwan, the first bDMARD, etanercept, was made available to patients via Taiwan's National Health Insurance (NHI) program in 2002. This marked the first time in Taiwan that a tumor necrosis factor-alpha (TNF-α) inhibitor had been used to treat active RA patients with inadequate response to methotrexate (MTX)-based conventional synthetic disease-modifying anti-rheumatic drugs (csDMARDs). Since then, a number of TNF-α and non-TNF-α inhibitors have also been approved. In December, 2014, the first Janus-kinase (Jak) inhibitor, tofacitinib, a tsDMARDs, was made available on Taiwan's NHI. TRACER enrolls RA patients from across Taiwan. Therefore, TRACER provides a great opportunity to investigate drug survival of bDMARDs and tsDMARDs for the treatment of RA.

In a prior systemic review and meta-analysis, better drug survival was found in etanercept-treated RA patients compared with two other TNF-α inhibitors [6]. However, concomitant use of csDMARDs, longer disease duration before initiation of a bDMARDs and female sex were associated with inferior drug survival [6]. Moreover, insufficient efficacy, adverse drug reactions, and safety signals of serious infections and malignancies could all contribute to biologic discontinuation [10–12]. In addition, shorter disease duration, baseline low disease activity, and young age may predict the 6-month therapeutic response in RA [9]. However, predictors of long-term drug retention for bDMARDs and tsDMARDs in RA are still lacking.

In the 2015 ACR guideline for the treatment of RA with high disease activity, bDMARDs were classified into TNF-α inhibitors and non-TNF-α biologics [2]. Although, rheumatoid factor (RF) and/or anti-citrullinated protein antibody (ACPA), especially at high levels, have been shown to be associated with erosive disease and poor outcome in RA, they have not consistently been shown to predict response to a variety of bDMARDs [3]. For example, RF and ACPA did not appear to be predictive of the response to anti-TNF treatment [13]. However, both ACPA and RF were previously found to predict a good EULAR response to rituximab therapy [14]. Moreover, in the AMPLE Trial, RA patients with the highest quartile of ACPA levels responded more favorably to abatacept, but not adalimumab, another anti-TNF-α

bDMARD [15]. Several studies did not find an association of seropositivity and responses to tocilizumab treatment [7]. It seems that RF and ACPA positivity may predict differential therapeutic responses in TNF-α inhibitors and non-TNF-α inhibitors. In addition, whether RF and ACPA are capable of predicting therapeutic responses in tofacitinib remains largely unknown.

To answer this unsolved question, this study aimed to identify predictors of 3-year drug retention for bDMARDs and tsDMARDs in a real-world dataset, TRACER.

## Materials and methods

### Data source

To identify predictors for drug survival of bDMARDs and tsDMARD, data were extracted from the Taiwan Rheumatology Association Clinical Electronic Registry (TRACER, www. tracer.org.tw), an investigator-led, Taiwan Rheumatology Association (TRA)-supported nationwide project. TRACER enrolled patients with autoimmune inflammatory rheumatic diseases, including RA, systemic lupus erythematosus, ankylosing spondylitis, and psoriatic diseases. This registry is a web-based system, which allows Taiwanese rheumatologists to register baseline demographic data, disease activity, autoantibodies status, medication, therapeutic responses, and adverse events of patients with systemic autoimmune diseases before and at 3-monthly intervals during treatment with csDMARDs, bDMARDs and tsDMARDs. TRA rheumatologists from tertiary referral hospitals, community hospitals, and local clinics voluntarily contributed de-linked patient information to this program.

### Study protocol

We conducted a 3-year drug survival study of bDMARDs and tsDMARDs in RA from the TRACER database. The diagnosis of established RA was done according to the ACR 1987 revised criteria and/or 2010 ACR/EULAR criteria for the classification of RA [16, 17]. Patients with active RA and who had started bDMARDs and tsDMARDs during Jan., 2007 to Aug., 2019 were eligible. They were inadequate responders to at least two csDMARDs including MTX and had a 28 joints-disease activity score (DAS28) >5.1 [18]. The baseline demographic data, the date of RA diagnosis, comorbidities, as well as serum levels of RF or ACPA before bDMARDs or tsDMARDs therapy were collected. All electronic procedures in TRACER and anonymized data are provided in S1 File. Participants with a follow-up period of less than 3 years were excluded. Taichung Veterans General Hospital's Ethics Committee approved the study (CE18190A), and waived the requirement for informed consent because the patients' data were anonymized prior to analysis.

### Treatment

Targeted therapies were classified as TNF-α bDMARDs (etanercept, adalimumab, golimumab), non-TNF-α bDMARDs (tocilizumab, abatacept, rituximab) and tsDMARDs (tofacitinib) treatment [2, 3]. In Taiwan, the reimbursement for bDMARDs and tsDMARDs in RA is only approved by NHI when a combination of MTX-based csDMARDs is prescribed. Therefore, these targeted therapies were administered in combination with csDMARDs unless participants were intolerant to MTX or csDMARDs.

### Study outcome

The primary outcome was 3-year drug retention. It was defined as treatment duration from the start date to discontinuation date of bDMARDs or tsDMARDs or the end of the

observation period plus one dispensation interval, whichever came first. The adverse events during bDMARDs and tsDMARDs treatment were recorded. The causes of bDMARDs and tsDMARDs discontinuation reflected treatment efficacy and adverse events ascertained by treating physicians.

### Covariates of interests

The baseline demographic data, disease duration, comorbidities, serum levels of RF or ACPA before bDMARDs or tsDMARDs therapy were extracted from TRACER. The RF IgM levels were measured by nephelometry (Dade Behring Inc., Newark, DE, USA, positive if $\geq$14 IU/mL). The ACPA levels were determined by EliA CCP (Phadia, Nieuwegein, The Netherlands, positive if $\geq$10 U/mL). Disease activity of RA was assessed by DAS28-ESR. Concomitant medications of glucocorticoids, csDMARDs, and previous exposure to bDMARDs or tsDMARDs were also recorded. Pre-existing comorbidities of hypertension, diabetes mellitus, cardiovascular disease, depression, and osteoporosis were obtained from medical records. Latent TB, hepatitis B carrier, and hepatitis C carrier statuses were identified following the risk management plan set forth by Taiwan's Centers for Disease Control (CDC) and TRA [19, 20].

### Patient and public involvement

We did not involve patients or the public in our work.

### Statistical analysis

The demographic data of the continuous parameters are shown as mean ± standard deviation, and for the categorical variables as the number of patients. Chi-Square test and Kruskal-Wallis test were used to compare variables among patients in the TNF-$\alpha$, non- TNF-$\alpha$, and tofacitinib groups. Risk factors associated with 3-year drug survival were determined by Cox proportional hazard regression. Statistically significant variables in univariate analyses were included in a multivariable model using the enter method. The drug retention probability curves were calculated by the Kaplan-Meier method, and statistical significance among groups was analyzed by the Log-rank test. All data were analyzed using the Statistical Package for the Social Sciences (SPSS) version 23.0. Significance was set at $p < 0.05$.

## Results

### Baseline demographic data

A total of 1,270 RA patients (TNF-$\alpha$ inhibitors: 584; non- TNF-$\alpha$ inhibitors: 535; tofacitinib: 151) were extracted from TRACER (Table 1). Of note, RA patients in the tofacitinib group exhibited older age (years, 58.1, 46.1–65.6 vs. 53.4, 41.6–60.8, and 57.3, 47.4–64.2, $p<$0.001), shorter disease duration (years, 9.5, 5.6–13.4 vs. 12.1, 8.3–13.4 and 13.1, 8.9–14.8, $p<$0.001), lower RF/ACPA positivity rates (RF positivity rates 73.3% vs. 99.5% and 80.4%, $p<$0.001; ACPA positivity rates 69.1% vs. 82.0% and 76.6%, $p$ = 0.003) and disease activity by DAS28-ESR (5.9, 5.2–6.5 vs. 6.4, 5.8–7.0, and 6.1, 5.5–6.6, $p<$0.001), lower glucocorticoid (mg per day, 5.0, 5.0–10.0 vs. 7.5, 5.0–10.0 and 7.5, 5.0–10.0, $p$ = 0.002), but higher MTX doses (mg per week, 15.0, 0.0–15.0 vs. 12.5, 5.0–15.0 and 10.0, 0.0–15.0, $p<$0.001) compared with the TNF-$\alpha$ inhibitors and non-TNF-$\alpha$ inhibitors groups (by Chi-square test or Kruskal-Wallis test, TNF-$\alpha$ inhibitors vs. tofacitinib, all $p<$0.01; non-TNF-$\alpha$ inhibitors vs. tofacitinib, all $p<$0.01).

**Table 1. Demographic data of RA patients receiving bDMARDs and tsDMARDs.**

| | TNF-α inhibitors (n = 584) | | Non-TNF-α inhibitors (n = 535) | | Tofacitinib (n = 151) | | *p* value |
|---|---|---|---|---|---|---|---|
| Age | 53.4 | (41.6–60.8) | 57.3 | (47.4–64.2) | 58.1 | (46.1–65.6) | <0.001** |
| Female gender | 491 | (84.1%) | 439 | (82.1%) | 131 | (86.8%) | 0.348 |
| Disease duration, years | 12.1 | (8.3–13.4) | 13.1 | (8.9–14.8) | 9.5 | (5.6–13.4) | <0.001** |
| RF positive | 581 | (99.5%) | 430 | (80.4%) | 110 | (73.3%) | <0.001** |
| ACPA positive | 466 | (82.0%) | 374 | (76.6%) | 85 | (69.1%) | 0.003** |
| ESR (mm/hr) | 44.0 | (28.0–64.0) | 42.0 | (26.0–71.0) | 37.0 | (24.0–55.0) | <0.001** |
| CRP (mg/dl) | 1.2 | (0.3–2.6) | 1.4 | (0.4–3.0) | 1.3 | (0.6–2.3) | 0.097* |
| DAS 28 | 6.4 | (5.8–7.0) | 6.1 | (5.5–6.6) | 5.9 | (5.2–6.5) | <0.001** |
| Tender joint count | 10.0 | (7.0–14.0) | 8.0 | (6.0–11.0) | 3.0 | (1.5–7.5) | <0.001** |
| Swelling joint count | 9.0 | (6.0–12.0) | 6.0 | (4.0–8.0) | 3.0 | (0.5–6.5) | <0.001** |
| Biologics-naive | 552 | (95.2%) | 237 | (45.1%) | 74 | (49.0%) | <0.001** |
| Hypertension | 115 | (19.7%) | 245 | (45.8%) | 38 | (25.2%) | <0.001** |
| Diabetes Mellitus | 37 | (6.3%) | 55 | (10.3%) | 23 | (15.2%) | 0.001** |
| Cardiovascular disease | 58 | (10.0%) | 53 | (9.9%) | 10 | (6.6%) | 0.427 |
| Depression | 6 | (1.0%) | 18 | (3.4%) | 10 | (6.6%) | <0.001** |
| Osteoporosis | 173 | (29.6%) | 206 | (38.8%) | 81 | (53.6%) | <0.001** |
| Latent TB | 174 | (30.4%) | 74 | (14.4%) | 14 | (9.8%) | <0.001** |
| HBV carrier | 39 | (6.7%) | 45 | (8.5%) | 10 | (6.6%) | 0.489 |
| HCV carrier | 22 | (3.8%) | 43 | (8.2%) | 9 | (6.0%) | 0.010* |
| Glucocorticoids dose (mg/day) | 7.5 | (5.0–10.0) | 7.5 | (5.0–10.0) | 5.0 | (5.0–10.0) | 0.002** |
| MTX dose (mg/week) | 12.5 | (5.0–15.0) | 10.0 | (0.0–15.0) | 15.0 | (0.0–15.0) | <0.001** |
| SAL | 359 | (62.0%) | 197 | (37.5%) | 70 | (46.4%) | <0.001** |
| HCQ | 429 | (74.1%) | 306 | (58.2%) | 102 | (67.5%) | <0.001** |
| LEF | 91 | (15.7%) | 124 | (23.6%) | 27 | (17.9%) | 0.004** |
| CsA | 101 | (17.4%) | 83 | (15.8%) | 16 | (10.6%) | 0.122 |

By Chi-square test or Kruskal-Wallis test.

*$p<0.05$

**$p<0.01$.

ACCP, anti-citrullinated protein antibody; CRP, C reactive protein; CSA, cyclosporine; DAS28-ESR, the 28 joints-erythrocyte sedimentation rate measurement; ESR, erythrocyte sedimentation rate; GI disease, gastrointestinal disease; HBV, hepatitis B virus; HCV hepatitis C virus; LEF, leflunomide; MTX, methotrexate; RF, rheumatoid factor; SAL, salazopyrin; TB, tuberculosis; TNF-α, tumor necrosis factor-alpha.

### Predictors associated with 3-year drug survival

To identify independent factors associated with 3-year drug retention, Cox regression analysis was performed (Table 2). We found that RF positivity (hazard ratio, HR: 0.48, 95% CI: 0.27–0.85, $p = 0.013$) and biologic-naïve status (HR: 0.61, 95% CI: 0.39–0.94, $p = 0.024$) were protective factors for drug retention. However, ACPA positivity (HR: 2.24, 95% CI: 1.32–3.79, $p = 0.003$) and latent TB infection (HR: 2.9, 95% CI: 2.06–4.09, $p<0.001$) were independent risk factors for drug discontinuation.

### Drug survival curves by biologics-exposure status

Among all participants (Fig 1A) and bio-naïve patients (Fig 1B), the TNF-α inhibitors group exhibited superior 3-year drug survival compared with the non-TNF-α inhibitors group (pairwise comparison, all $p<0.001$). However, drug retention rates seemed comparable in biologics-experienced patients (Fig 1C, pairwise comparison, all $p>0.05$).

**Table 2. Cox regression analysis of factors associated with 3-year drug survival in RA patients receiving bDMARDs and tsDMARDs treatment.**

| | Univariate | | | Multivariable | | |
|---|---|---|---|---|---|---|
| | **HR** | **95%CI** | | *p* **value** | **HR** | **95%CI** | | *p* **value** |
| Age | | | | | | | |
| <65y | Reference | Reference | | | Reference | Reference | | |
| ≥65y | 1.54 | (1.18- | 2.01) | 0.002** | 0.96 | (0.64- | 1.42) | 0.830 |
| Gender | | | | | | | |
| F | Reference | Reference | | | | | |
| M | 0.83 | (0.59- | 1.16) | 0.272 | | | |
| Disease duration, years | 0.99 | (0.97- | 1.01) | 0.302 | | | |
| RF positive | 0.66 | (0.48- | 0.92) | 0.015* | 0.48 | (0.27- | 0.85) | 0.013* |
| ACPA positive | 1.58 | (1.13- | 2.23) | 0.008** | 2.24 | (1.32- | 3.79) | 0.003** |
| ESR | 1.00 | (1.00- | 1.01) | 0.162 | | | |
| CRP | 1.00 | (0.99- | 1.00) | 0.718 | | | |
| DAS 28 | 1.00 | (0.96- | 1.05) | 0.924 | | | |
| Biologics-naive | 0.62 | (0.48- | 0.79) | <0.001** | 0.61 | (0.39- | 0.94) | 0.024* |
| Hypertension | 1.27 | (0.99- | 1.62) | 0.056 | | | |
| Diabetes Mellitus | 1.55 | (1.09- | 2.21) | 0.016* | 1.52 | (0.94- | 2.45) | 0.089 |
| Cardiovascular disease | 1.34 | (0.94- | 1.92) | 0.107 | | | |
| Depression | 1.82 | (1.02- | 3.25) | 0.042* | 1.47 | (0.52- | 4.14) | 0.462 |
| Osteoporosis | 1.47 | (1.16- | 1.86) | 0.002** | 0.86 | (0.62- | 1.20) | 0.382 |
| Latent TB | 2.05 | (1.59- | 2.64) | <0.001** | 2.90 | (2.06- | 4.09) | <0.001** |
| HBV carrier | 0.91 | (0.57- | 1.45) | 0.693 | | | |
| HCV carrier | 2.39 | (1.62- | 3.51) | <0.001** | 1.17 | (0.55- | 2.47) | 0.679 |
| bDMARDs & tsDMARDs | | | | | | | |
| TNF-α inhibitors | Reference | Reference | | | Reference | Reference | | |
| Non-TNF-α inhibitors | 1.73 | (1.35- | 2.23) | <0.001** | 0.78 | (0.49- | 1.25) | 0.301 |
| Tofacitinib | 1.39 | (0.92- | 2.12) | 0.121 | 0.65 | (0.15- | 2.85) | 0.566 |
| Glucocorticoids dose | 0.99 | (0.97- | 1.02) | 0.665 | | | |
| MTX dose | 0.98 | (0.96- | 0.99) | 0.008** | 1.00 | (0.97- | 1.03) | 0.942 |
| SAL | 0.83 | (0.65- | 1.05) | 0.116 | | | |
| HCQ | 0.91 | (0.71- | 1.17) | 0.478 | | | |
| LEF | 1.44 | (1.09- | 1.90) | 0.009** | 1.38 | (0.90- | 2.13) | 0.141 |
| CsA | 1.08 | (0.78- | 1.48) | 0.644 | | | |

By Cox proportional hazard regression.

*p<0.05

**p<0.01.

ACCP, anti-citrullinated protein antibody; CRP, C reactive protein; CSA, cyclosporine; DAS28-ESR, the 28 joints-erythrocyte sedimentation rate measurement; ESR, erythrocyte sedimentation rate; GI disease, gastrointestinal disease; HBV, hepatitis B virus; HCV hepatitis C virus; LEF, leflunomide; MTX, methotrexate; n, number of patients included in analysis; RF, rheumatoid factor; SAL, salazopyrin; TB, tuberculosis; TNF-α, tumor necrosis factor-alpha.

## Drug survival curves by causes of discontinuation

Fig 2 displays the causes of drug discontinuation in treatment groups. In Fig 2A, it can be seen that patients taking TNF-α blockers were at lower risk for discontinuation due to inefficacy compared with those in the non-TNF-α blockers and tofacitinib groups (pairwise comparison, TNF-α inhibitors vs. non-TNF-α inhibitors, *p*<0.001; TNF-α inhibitors vs. tofacitinib, *p* = 0.001; non-TNF-α inhibitors vs. tofacitinib, *p*>0.05). Moreover, the tofacitinib and non-TNF-α inhibitors groups were at higher risk of discontinuation due to adverse events (Fig 2B,

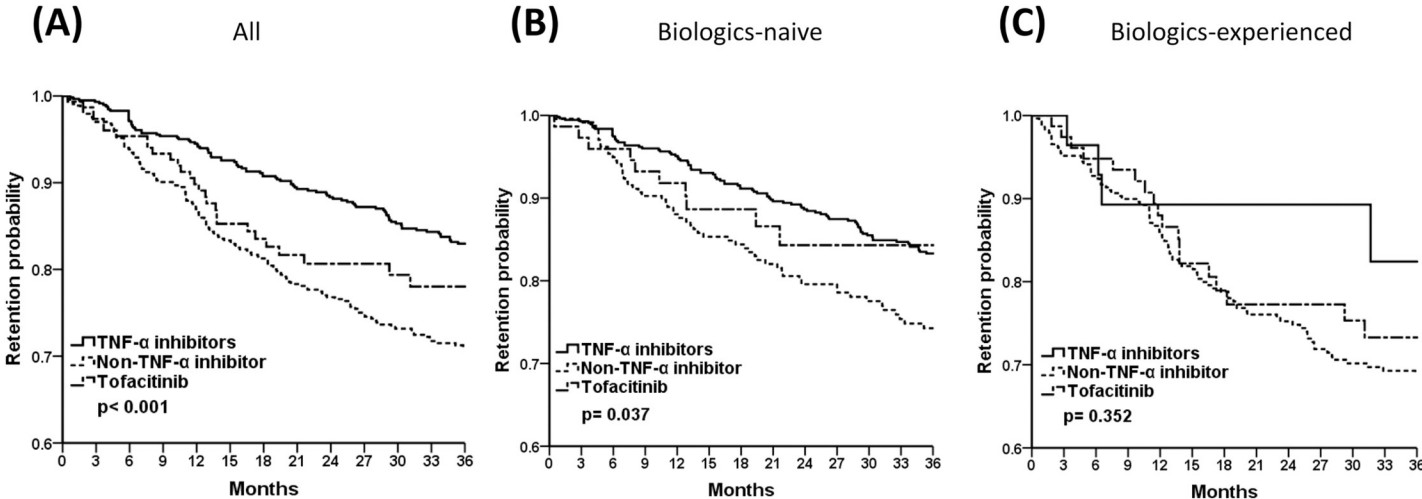

**Fig 1.** The 3-year drug retention probability of TNF-α inhibitors, non-TNF-α inhibitors, and tofacitinib in (A) all, (B) biologics-naïve, (C) biologics-experienced RA patients by Kaplan-Meier survival analysis. Pairwise comparison (A) all *p*<0.001, (B) TNF-α inhibitors vs. non-TNF-α inhibitors, *p* = 0.010; TNF-α inhibitors vs. tofacitinib & non-TNF-α inhibitors vs. tofacitinib, *p*>0.05 (C) all *p*>0.05.

pairwise comparison, TNF-α inhibitors vs. non-TNF-α inhibitors, *p* = 0.014; TNF-α inhibitors vs. tofacitinib & non-TNF-α inhibitors vs. tofacitinib, *p*<0.001).

## Individual drug survival curves by RF and ACPA positivity

Since RF and ACPA were independent factors associated with drug survival, we examined the seropositivity status and drug retention in various treatment groups (Fig 3). In the TNF-α inhibitors and tofacitinib groups, we found that ACPA negativity was associated with a better 3-year drug retention probability (Fig 3B and 3J), ACPA (+) vs. ACPA (-) in TNF-α inhibitors, *p*<0.001 and tofacitinib, *p* = 0.025). In contrast, RF and ACPA positivity predicted better drug

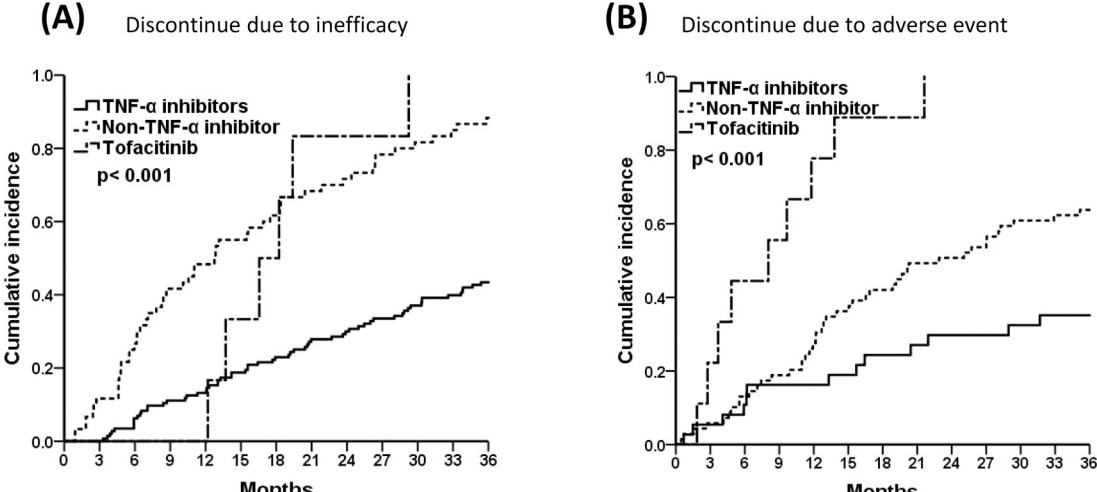

**Fig 2.** The 3-year cumulative incidences of TNF-α inhibitors, non-TNF-α inhibitors, and tofacitinib, (A) Discontinued due to inefficacy, and (B) Discontinued due to adverse event by Kaplan-Meier survival analysis. Pairwise comparison (A) TNF-α inhibitors vs. non-TNF-α inhibitors, *p*<0.001; TNF-α inhibitors vs. tofacitinib, *p* = 0.001; non-TNF-α inhibitors vs. tofacitinib, *p*>0.05, (B) TNF-α inhibitors vs. non-TNF-α inhibitors, *p* = 0.014; TNF-α inhibitors vs. tofacitinib & non-TNF-α inhibitors vs. tofacitinib, *p*<0.001.

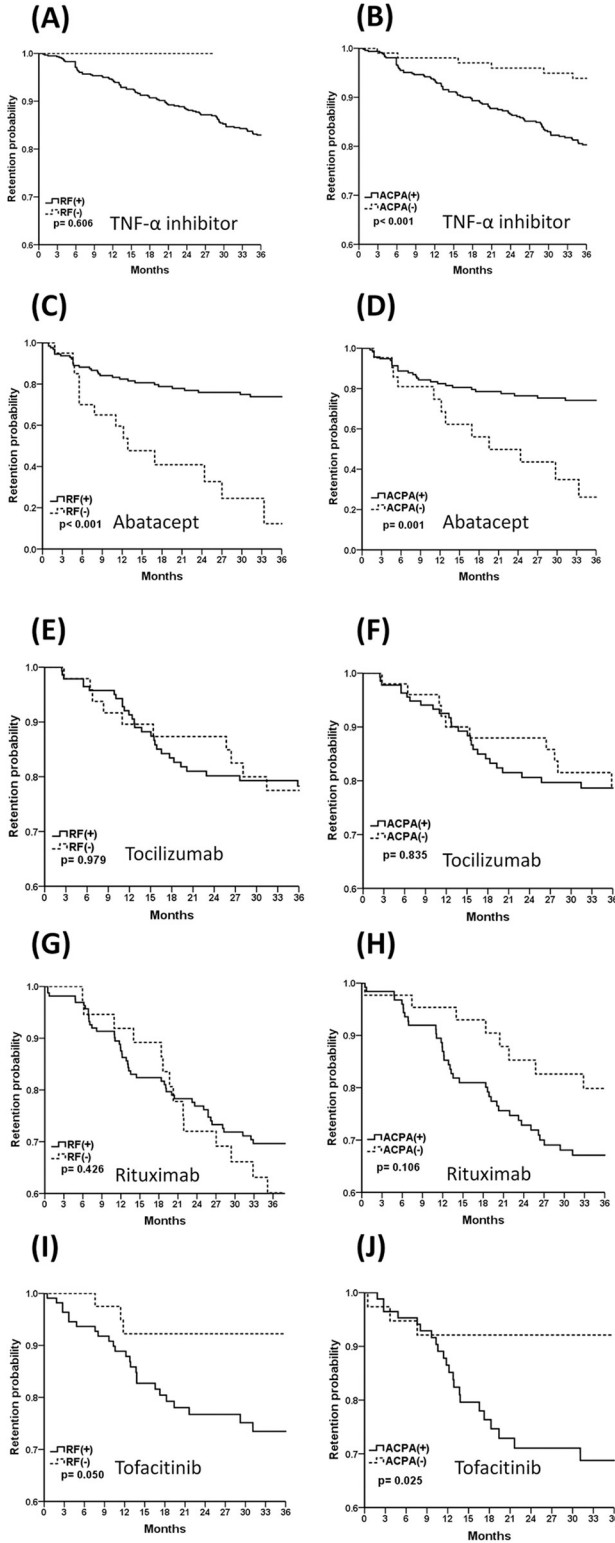

**Fig 3.** The 3-year Kaplan-Meier drug retention probability of TNF-α inhibitors (A, B), abatacept (C, D), tocilizumab (E,F), rituximab (G, H), and tofacitinib (I, J) by RF and ACPA positivity. TNF-α, tumor necrosis factor-alpha; RF, rheumatoid factor; ACPA, anti-citrullinated protein antibody.

survival in abatacept-treated patients (Fig 3C and 3D, RF(+) vs. RF (-), $p<0.001$ and ACPA(+) vs. ACPA(-), $p = 0.001$). Interestingly, seropositivity appeared to have no impact on drug retention in the tocilizumab and rituximab groups (Fig 3E–3H, all $p> 0.05$).

## Discussion

In this study, we aimed to investigate the predictors of 3-year drug survival for bDMARDs and tsDMARDs in RA using a real-world dataset in Taiwan. We demonstrated that TNF-α inhibitors, non-TNF-α inhibitors, and tofacitinib appeared to have differential long-term drug retention rates. Our study showed that bio-naïve status predicted better drug survival in TNF-α inhibitors-treated RA patients. However, concomitant latent TB infection predicted drug discontinuation. RF and ACPA positivity predicted better abatacept drug survival. In contrast, ACPA negativity was associated with superior TNF-α inhibitors and tofacitinib survival. The findings presented herein are the first to demonstrate that seropositivity seems to be a potential predictor for drug survival of bDMARDs and tsDMARDs in RA. This novel finding might shed light on the use of RF and ACPA as biomarkers for precision treatment and drug retention in RA.

Our results indicate that RA patients who received TNF-α inhibitors exhibited better 3-year drug retention compared with their counterparts. This could be related to the fact that etanercept and adalimumab were the first two bDMARDs approved in Taiwan. Therefore, a high proportion (95.2%) of the TNF-α antagonist group comprised bio-naïve patients from TRACER. We also showed that bio-naïve status was an independent and favorable factor for drug survival. This is consistent with previously reported drug persistence rates of TNF-α inhibitors, which indicate they were better as a first-line biologic agent than as a second-line therapy for RA [8]. A greater proportion of anti-TNF-treated patients in our cohort were bDMARD-naïve patients, suggesting they might have a more rapid reduction in disease activity and greater improvements in physical function related to active RA in comparison with bDMARD-experienced patients [21].

In the present study, the concomitant latent TB infection was an independent risk predictor for drug discontinuation. Taiwan is an endemic area of latent TB infection and in over 80% of latent TB cases, a prophylaxis strategy was applied [22]. We previously demonstrated that the 1-year TB risk in RA patients receiving TNF-α inhibitors was higher than that found among patients receiving non-TNF-α inhibitors in a nationwide population-based study between 2008 and 2012 [23]. An Italian study also demonstrated that RA patients with latent TB might discontinue anti-TNF therapy, because active TB occurred during and after anti-TB prophylactic therapy [24]. The disruption of granuloma integrity by anti-TNF therapy contributes to increased risk of latent TB reactivation [25]. Anti-TB prophylaxis could reduce TB reactivation by 65% [26]. Screening and prophylaxis of latent TB was advocated in 2012, and since then the incidence of TB infection has decreased. Moreover, the 5-year cumulative TB incidence between TNF-α inhibitors and non-TNF-α were indistinguishable [27]. Since our study enrolled participants before the era of universal screening and prophylaxis for latent TB, we found that it remained a risk factor for bDMARDs and tsDMARDs discontinuation.

Our study showed that patients with RA under anti-TNF treatment were less likely to discontinue targeted therapies due to inefficacy and adverse events. Immunogenicity to monoclonal antibodies of TNF-inhibitors could lead to the formation of anti-drug antibodies, and was correlated with loss of treatment efficacy [11, 28, 29]. However, non-TNF inhibitors were less frequently associated with immunogenicity [30]. Moreover, almost all RA under bDMARDs and tsDMARDs treatment in this study were combined with MTX-based csDMARDs, and thus our results might have been biased toward overestimation of the drug retention rates of

TNF-α inhibitors [2, 3]. A small molecule tsDMARD, tofacitinib, has not been reported to elicit immunogenicity. Nevertheless, our data showed that discontinuation of non-TNF inhibitors and tofacitinib may occur after 1 year of treatment. Further studies are needed to elucidate the mechanisms whereby patients may lose clinical responses to non-TNF inhibitors and tofacitinib.

The 3-year observational study showed that the tofacitinib and non-TNF-α inhibitors groups had a higher risk of discontinuation because of adverse events compared with the TNF-α inhibitors group. Accordingly, the adverse events for discontinuation of bDMARDs and tsDMARDs were mainly infection and drug intolerance especially in elderly patients with RA [10, 12]. A prior observational study reported that abatacept was associated with lower incidence rates of serious infections and severe infusion/injection reactions [31], as well as higher drug retention rates compared with other biologic agents [12]. Meanwhile, tocilizumab exhibited similar risk of opportunistic and bacterial infection but lower TB reactivation in comparison to anti-TNF therapy [32]. However, the prior anti-TNF experience was also associated with increased incidence of infection and serious infection during tocilizumab therapy [32, 33]. The same phenomenon was observed for B cell depletion agent, with a relatively high prevalence of severe pneumonia, and reactivation of HBV hepatitis and herpes zoster [34]. Previous reports also showed that the overall risk of infection, serious infection, and mortality rates in RA with tofacitinib were similar to those of other bDMARDs [35, 36], with a particular safety signal in herpes zoster infection especially in those receiving glucocorticoids [37]. Since the RA patients under anti-TNF treatment were younger and had a higher proportion of bionaïve status, we cannot exclude the potential effects of confounders on the risk of discontinuation by adverse events.

Our study showed that RF positivity predicted a better 3-year drug survival in RA. Accordingly, seropositive RA patients share specific genetic and environment risk factors which differ from seronegative RA in clinical course and prognosis [38]. The autoantibodies of RF and ACPA in RA have direct pathogenic contributions to disease progression and seem to be a useful biomarker as a clinical predictor of drug survival [15, 39]. Systematic reviews indicate that neither RF nor ACPA status in RA patients is a predictor associated with response to TNF-α inhibitors [15, 40, 41]. However, our result demonstrated that the absence of baseline ACPA was associated with better drug survival of TNF-α inhibitors. Previous studies showed that high serum levels of RF were associated with poor treatment response of TNF-α inhibitors [42, 43]. Since RF- and ACPA-negative RA typically showed less bony erosion and structural damage, and exhibited a modest disease course [38], we speculate these patients might respond better to TNF-α inhibitors with less discontinuation due to inefficacy, leading to better drug retention rates.

Our result demonstrated that RF- and ACPA-positive RA patients receiving abatacept treatment exhibited better drug survival compared with the seronegative group. Inhibition of T cell co-stimulation factors may selectively affect autoantibody production [44, 45]. This was in keeping with the post-hoc analysis of the AMPLE study demonstrating that in RA patients with highest baseline anti-CCP2 antibody concentrations there was a stronger correlation with better clinical response to abatacept compared with those with lower concentrations [15]. Meanwhile, an international, prospective real-world study also showed that both RF and ACPA positivity were associated with higher abatacept retention [46]. Taken together, seropositivity of RF and ACPA could predict the likelihood of drug retention of T cell co-stimulation inhibitors.

A previous systematic review and meta-analysis showed that the baseline RF positivity in RA patients predicts better response to rituximab and tocilizumab [47]. However, a large cohort study showed contradictory results, i.e., neither baseline RF nor ACPA was a predictor of better response for tocilizumab therapy in RA [7]. Our results suggest that neither RF nor

ACPA status was a good predictor of drug survival of tocilizumab or rituximab. In Taiwan, rituximab is only approved for biologic-experienced RA patients; tocilizumab was also available on the NHI program for second-line therapy for the first 3 years that it was on the market in Taiwan. Since biologic-experienced RA patients tend to respond less favorably to therapy, our result may not be extrapolated to bio-naïve patients.

The post-hoc analysis of tofacitinib treatment indicates that treatment outcome is not markedly influenced by autoantibody seropositivity [48]. Moreover, a higher proportion of tofacitinib-treated seropositive RA patients exhibited ACR20/50/70 responses, low disease activity, and remission in comparison with seronegative RA, especially with 10mg two times a day. Surprisingly, our study suggested that seronegative RA was associated with better tofacitinib drug retention. Since tofacitinib, a JAK1 and JAK3 kinase inhibitor, targets multiple cytokine receptors and exhibits diverse in vitro effects [49, 50], we postulate that seronegative RA patients might share a common pathology with the JAK pathway that could be targeted by tofacitinib. Further study is needed to confirm our finding and elucidate the underlying mechanisms.

Our study did have several limitations. First, the baseline characteristics among different treatment groups were not equally distributed. Although Cox regression analysis was performed to analyze independent factors associated with drug survival, there may have been confounding factors that were not completely controlled for. However, this also reflected the prescription behavior of rheumatologists in real-world observational studies. Second, various bDMARDs and tsDMARDs appeared on the market across a period spanning more than 10 years. In the early years, when only two TNF-α inhibitors, etanercept and adalimumab, were available, the limited choice of bDMARDs may have predisposed the treated patients to stay on these drugs. This might have led to an over-estimation of drug retention rates of TNF-α inhibitors. However, our study analyzed data from a period of more than a decade, which may provide long-term evidence of drug survival of bDMARDs and tsDMARDs. Third, the causes of drug discontinuation were diverse. We did not re-classify RA patients whose targeted therapies showed poor efficacy into primary and secondary treatment failure. Moreover, adverse events also included multiple diverse causes. A large inception cohort is needed to investigate the precise causes of drug discontinuation.

In conclusion, this long-term real-world study using the TRACER database demonstrated that bio-naïve status, latent TB infection, and RF/ACPA seropositivity were potential predictors for 3-year drug retention of bDMARDs and tsDMARDs in RA. Bio-naïve status was associated with better drug survival in TNF-α inhibitor-treated RA patients. RF and ACPA positivity seems to predict better abatacept drug retention. Conversely, ACPA-negative RA patients appeared to tolerate TNF-α inhibitors and tofacitinib therapies better than their counterparts. Thus, clinical parameters and autoantibody status may be used as a potential guide for targeted therapy in RA patients.

## Supporting information

**S1 File.**
(XLSX)

**S2 File.**
(PDF)

## Acknowledgments

The authors thank Mr. Jun-Peng Chen at the Biostatistics Task Force of Taichung Veterans General Hospital for assisting with the statistical analysis.

## Author Contributions

**Conceptualization:** Ching-Tsai Lin, Yi-Ming Chen.

**Data curation:** Ching-Tsai Lin, Yi-Ming Chen.

**Formal analysis:** Ching-Tsai Lin, Wen-Nan Huang, Wen-Chan Tsai, Jun-Peng Chen, Wei-Ting Hung, Tsu-Yi Hsieh, Hsin-Hua Chen, Chia-Wei Hsieh, Kuo-Lung Lai, Kuo-Tung Tang, Chih-Wei Tseng, Der-Yuan Chen, Yi-Hsin Chen, Yi-Ming Chen.

**Funding acquisition:** Yi-Ming Chen.

**Methodology:** Jun-Peng Chen.

**Writing – original draft:** Ching-Tsai Lin.

**Writing – review & editing:** Wen-Nan Huang, Wen-Chan Tsai, Jun-Peng Chen, Wei-Ting Hung, Tsu-Yi Hsieh, Hsin-Hua Chen, Chia-Wei Hsieh, Kuo-Lung Lai, Kuo-Tung Tang, Chih-Wei Tseng, Der-Yuan Chen, Yi-Hsin Chen, Yi-Ming Chen.

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
