## [Decision Letter · Decision Letter 0]

8 Feb 2021

PONE-D-20-23397

Seropositivity Associated with Drug Survival of Biologic and Targeted Synthetic DMARDs in Rheumatoid Arthritis: Analysis from the TRA Clinical Electronic Registry

PLOS ONE

Dear Dr. Chen,

Thank you for submitting your manuscript to PLOS ONE. After careful consideration, we feel that it has merit but does not fully meet PLOS ONE’s publication criteria as it currently stands. Therefore, we invite you to submit a revised version of the manuscript that addresses the points raised during the review process.

We look forward to receiving your revised manuscript.

Kind regards,

Vineet Kumar Rai, PhD

Academic Editor

PLOS ONE

Journal Requirements:

2. Please refer to any post-hoc corrections to correct for multiple comparisons during your statistical analyses. If these were not performed please justify the reasons. Please refer to our statistical reporting guidelines for assistance (https://journals.plos.org/plosone/s/submission-guidelines.#loc-statistical-reporting).

3.We note that you have indicated that data from this study are available upon request. PLOS only allows data to be available upon request if there are legal or ethical restrictions on sharing data publicly. For information on unacceptable data access restrictions, please see http://journals.plos.org/plosone/s/data-availability#loc-unacceptable-data-access-restrictions.

Reviewers' comments:

Reviewer's Responses to Questions

**Comments to the Author**

1. Is the manuscript technically sound, and do the data support the conclusions?

Reviewer #1: Yes

Reviewer #2: Yes

Reviewer #3: No

2. Has the statistical analysis been performed appropriately and rigorously? 

Reviewer #1: Yes

Reviewer #2: No

Reviewer #3: No

3. Have the authors made all data underlying the findings in their manuscript fully available?

Reviewer #1: Yes

Reviewer #2: Yes

Reviewer #3: Yes

4. Is the manuscript presented in an intelligible fashion and written in standard English?

Reviewer #1: Yes

Reviewer #2: Yes

Reviewer #3: No

5. Review Comments to the Author

Reviewer #1: Comment to Author

This is an interesting study and the authors have collected well structured data. The paper is generally well written and structured. could be improved rephrasing the summary of outcome with objective under study.

Reviewer #2: in this study, researchers found that RA patients who received TNF-α inhibitors exhibited significantly  better drug retention compared with their counterparts. This could be related to the fact that etanercept and adalimumab were the first two bDMARDs approved in Taiwan. Therefore, a high proportion (95.2%) of the TNF-α antagonist group comprised bio-naïve patients. The researchers showed that bio-naïve status was an independent and favourable factor for drug survival. This is consistent with previously reported drug persistent rates of TNF-α inhibitors which indicate they were better as a first-line biologic agent than as a second-line therapy for RA . A greater proportion of anti-TNF-treated patients in our cohort were bDMARD-naïve patients, suggesting they might have a more rapid reduction in disease activity and greater improvements in physical function related to active RA in comparison with bDMARD-experienced patients .

The research findings are justifying the objective of this project. Kindly check the grammatical errors and resubmit.

Reviewer #3: Dear Author,

Thanks for submitting your research manuscript entitled “Seropositivity Associated with Drug Survival of Biologic and Targeted Synthetic DMARDs in Rheumatoid Arthritis: Analysis from the TRA Clinical Electronic Registry”

Major concerns

Please find out the following comments

The rationale for the study is not mentioned or incomplete regarding Targeted Synthetic DMARDs in Rheumatoid Arthritis.

The rationale behind the Analysis from the TRA Clinical Electronic Registry is also not clear and should be specified in the introduction and discussed in the results.

Overall the whole manuscript must be again thoroughly revised to correct grammar, punctuation, and syntax. The authors should use small bites of sentences to tell their views or literature review in a comprehensive way, rather than using complicated and hectic lines and sentences.

Title:

The author should have revised the manuscript's title concerning the core concept of the research. More like a statement to the reviewer, the title does not display any clear claim to the study regarding drug Survival of Biologic and Targeted Synthetic DMARDs in Rheumatoid Arthritis.

Abstract:

- The rationale behind this research is not well explained, and several major concerns still constrain the reviewer's enthusiasm for publishing this manuscript.

- The abstract has no specific information about the study schedules and creates uncertainty

- The author failed to offer a straightforward conclusion to the abstract and should be more descriptive and follow the research’s rationale.

Introduction:

- In addition, the basic literature is not well written and does not even include any literature on alternative approaches in the treatment of Rheumatoid Arthritis.

- What were the studies in existing research that were included? These are the basic questions which the author needs to address.

- The authors should present the justification behind the selection of o three major groups: tumor necrosis factor-alpha (TNF-α) inhibitors, non-TNF-α inhibitors, and tofacitinib.

- In the intro, the author should have furnished the individual references for these.

- The statement and conclusive remarks provided in the introduction are confusing and not matched with the clear observation associated with abstract, results and discussion parts.

Material and methods:

- The author should have given the rationale or study schedule behind the research. The author does not mention any specific protocol in a selection of data source and study population.

- What are the altered results related to the combination treatment? The author must provide answers to all these types of questions.

- The M&M writing pattern is not made scientifically, creating confusion through the format of disorganized writing regarding the electronic registry.

- In order to support the assessment of all mentioned parameters in his study, the author should provide all the source documents and data he has followed for all electronic procedures.

Results:

- Results need more clarification and significant justification. Differentiating between the outcome and the discussion sections is quite difficult. It looks like; there is no scientific clarity, and follow continuous paragraph writing without any point discussion.

- Authors fail to articulate the outcomes properly in Baseline demographic data, Factors associated with 3-year drug survival, Drug survival curves by biologics-exposure status, Drug survival curves by causes of discontinuation, and Individual drug survival curves by RF and ACPA positivity. Therefore, the findings should be cautiously revised.

- The author should strengthen and organize the result section more significantly and scientifically.

- The author should provide the raw data in scanned PDF or excel format (including institutional certificates of ethical approval, practical record register, practical record entry, or specific instrument/equipment data entry).

Discussion:

- Significant scientific statements and observations are without proper reference in the discussion. When explaining the outcomes, the author needs to be very cautious.

- Author's needs to orient and provide the scientific justification to correlate all the parameters he performed, in a very scientific and significant manner.

- In both the discussion and the conclusion, the aims, rationale, and future perspectives are not evident clearly.

- The discussion is usually organized at the beginning to address all the observations and evaluate them at the end. It makes the results easier to contextualize and simpler to comprehend. Furthermore, a minimal critical analysis should be provided.

- According to the reviewer, a more thorough analysis of the findings is needed in the view of the current literature. To address the outcome measures/results separately and how they correlate with the existing literature, it would be better if the author restructured to take a more critical approach.

- The reviewer feels that the conclusion part is somewhere that seems out of focus.

- The reviewer believes that the conclusion section appears to be out of sight somewhere.

6. PLOS authors have the option to publish the peer review history of their article (what does this mean?). If published, this will include your full peer review and any attached files.

Reviewer #1: No

Reviewer #2: No

Reviewer #3: **Yes: **Sidharth Mehan

---

## [Author Response · Author response to Decision Letter 0]

3 Apr 2021

Reply to the comments and suggestions from Reviewer 1

Thanks for your excellent review and useful comments

This is an interesting study and the authors have collected well structured data. The paper is generally well written and structured. could be improved rephrasing the summary of outcome with objective under study.

Thanks for the reviewer’s comments. We revsied the objective and outome of this study in the abstract (lines 7-8) and introduction section (page 7, lines 6-7).

 

Reply to the comments and suggestions from Reviewer 2

Thanks for your excellent review and useful comments

 In this study, researchers found that RA patients who received TNF-α inhibitors exhibited significantly better drug retention compared with their counterparts. This could be related to the fact that etanercept and adalimumab were the first two bDMARDs approved in Taiwan. Therefore, a high proportion (95.2%) of the TNF-α antagonist group comprised bio-naïve patients. The researchers showed that bio-naïve status was an independent and favourable factor for drug survival. This is consistent with previously reported drug persistent rates of TNF-α inhibitors which indicate they were better as a first-line biologic agent than as a second-line therapy for RA . A greater proportion of anti-TNF-treated patients in our cohort were bDMARD-naïve patients, suggesting they might have a more rapid reduction in disease activity and greater improvements in physical function related to active RA in comparison with bDMARD-experienced patients .

The research findings are justifying the objective of this project. Kindly check the grammatical errors and resubmit.

Thanks for the reviewer’s suggestion. We have sent the manuscript for English editing to correct grammatical errors. 

Reply to the comments and suggestions from Reviewer 3

Thanks for your excellent review and useful comments

Major concerns

Please find out the following comments

The rationale for the study is not mentioned or incomplete regarding Targeted Synthetic DMARDs in Rheumatoid Arthritis. 

We appreciate the reviewer’s comments; we have added this to our abstract and introduction. We have revised our manuscript on page 3, lines 2-5, and 7-10; page 5, lines 3-7, and 10-12; page 6, lines 1-11; and page 7, lines 2-7.

The rationale behind the Analysis from the TRA Clinical Electronic Registry is also not clear and should be specified in the introduction and discussed in the results. Data Sauce of TRA Clinical Electronic Registry (TRACER)

Thanks for this valuable comment; we have specified the rationale behind the analysis from the TRACER in the introduction and discussion sections. We have revised our manuscript on page 5, lines 13-19; page 6, lines 1-11; and page 7, lines 2-7.

Overall the whole manuscript must be again thoroughly revised to correct grammar, punctuation, and syntax. The authors should use small bites of sentences to tell their views or literature review in a comprehensive way, rather than using complicated and hectic lines and sentences.

We have sent this manuscript for English editing to correct grammar, punctuation, and syntax, according to your recommendation. We have also revised our manuscript in a comprehensive way so that readers can more easily understand this paper. 

Title: 

The author should have revised the manuscript's title concerning the core concept of the research. More like a statement to the reviewer, the title does not display any clear claim to the study regarding drug Survival of Biologic and Targeted Synthetic DMARDs in Rheumatoid Arthritis.

As per your suggestion, we have revised the manuscript’s title as follows: “Predictors of Drug Survival for Biologic and Targeted Synthetic DMARDs in Rheumatoid Arthritis: Analysis from the TRA Clinical Electronic Registry”

Abstract: 

- The rationale behind this research is not well explained, and several major concerns still constrain the reviewer's enthusiasm for publishing this manuscript.

- The abstract has no specific information about the study schedules and creates uncertainty.

We appreciate the reviewer’s comments. We have added our study schedule for this study and have revised our abstract. Please see page 3, lines 4-10.

- The author failed to offer a straightforward conclusion to the abstract and should be more descriptive and follow the research’s rationale.

Thanks for this useful comment, we have revised the conclusion of the abstract, accordingly. Please see page 3, lines 21-23.

Introduction:

- In addition, the basic literature is not well written and does not even include any literature on alternative approaches in the treatment of Rheumatoid Arthritis.

Thanks again for your valuable comment; we have added ACR/EULAR recommendations for the management of RA to include alternative approaches and we have revised our introduction. Please see page 5, lines 3-7.

- What were the studies in existing research that were included? These are the basic questions which the author needs to address.

Thank you for noting this. We have revised our manuscript accordingly and the introduction now includes a review of drug-survival studies on page 6, lines 4-11.

- The authors should present the justification behind the selection of three major groups: tumor necrosis factor-alpha (TNF-α) inhibitors, non-TNF-α inhibitors, and tofacitinib.

In the intro, the author should have furnished the individual references for these.

Thanks for your suggestion. The bDMARDs were classified into TNF-α inhibitors and non-TNF-α inhibitors according to their various mechanisms and by the ACR/eular treatment recommendations. We have provided references related to how RF and ACPA positivity may predict differential therapeutic responses in TNF-α inhibitors and non-TNF-α inhibitors. We have revised the introduction to explain the selection of these 3 major groups. Please see page 6, lines 12-19; and page7, lines 1-7. 

- The statement and conclusive remarks provided in the introduction are confusing and not matched with the clear observation associated with abstract, results and discussion parts.

We appreciate the reviewer’s comments; we have revised our concluding remarks in the introduction, accordingly. Please see page 7, lines 2-7.

Material and methods:

- The author should have given the rationale or study schedule behind the research. The author does not mention any specific protocol in a selection of data source and study population. 

Thank you for your useful comments. We have added the rationale behind the study on page 8, lines 3-5, and the study schedule is presented on page 8, lines 5-13 in the revised manuscript. We also added a subheading of the study protocol in the methods section on page 8, lines 14-19; and page 9, lines 1-5.

- What are the altered results related to the combination treatment? The author must provide answers to all these types of questions.

We appreciate the reviewer’s comments. In Taiwan, the reimbursement for bDMARDs and tsDMARDs in RA is approved by Taiwan’s National Health Insurance program only when a combination of MTX-based csDMARDs is prescribed. As per the ACR/EULAR recommendations for management of RA, combination therapy for all bDMARDs and tsDMARDs demonstrates superior drug response and structural efficacy. Moreover, the majority of patients in this cohort received combination therapies with MTX-based csDMARDs. It has been reported that MTX could reduce the incidence of immunogenicity of TNF-α inhibitors. Our result might have been biased toward overestimation of the drug retention rates of TNF-α inhibitors. We have revised our manuscript accordingly on page 9, lines 8-12; page 18, lines 17-19; and page19, lines 1-3.

- The M&M writing pattern is not made scientifically, creating confusion through the format of disorganized writing regarding the electronic registry. 

Thanks for the reviewer’s valuable input. We have reorganized the description of our data source (page 8, line 3-12) and study protocol (page 8, lines 14-19; and page 9, lines 1-5) to provide a better understanding of the electronic registry, TRACER.

- In order to support the assessment of all mentioned parameters in his study, the author should provide all the source documents and data he has followed for all electronic procedures.

Thanks for your suggestions. We have provided source documents for all electronic procedures in TRACER and anonymized data in the supplementary file 1. Please see page 9, lines 1-2. 

Results:

- Results need more clarification and significant justification. Differentiating between the outcome and the discussion sections is quite difficult. It looks like; there is no scientific clarity, and follow continuous paragraph writing without any point discussion.

Thanks for the reviewer’s suggestions. We have revised our results section thoroughly on pages 12 – 16.

- Authors fail to articulate the outcomes properly in Baseline demographic data, Factors associated with 3-year drug survival, Drug survival curves by biologics-exposure status, Drug survival curves by causes of discontinuation, and Individual drug survival curves by RF and ACPA positivity. Therefore, the findings should be cautiously revised.

We have revised our results section to provide greater scientific clarity in accordance with your recommendation. Please see page 12, lines 5-12; page 15, line 4; page 15 lines 8-10, 11-13, 18-19; and page 16, lines 1-3.

- The author should strengthen and organize the result section more significantly and scientifically.

We appreciate your useful comments. We have revised the subheadings and contents of the results section.Please see page 12, lines 5-12; page 15, line 4; page 15 lines 8-10, 11-13, 18-19; and page 16, lines 1-3. 

- The author should provide the raw data in scanned PDF or excel format (including institutional certificates of ethical approval, practical record register, practical record entry, or specific instrument/equipment data entry).

As per the reviewer’s suggestion, we have provided raw data, institutional certificates of ethical approval and record register(pdf format) in the supplementary files 1, 2 & 3.

Discussion:

- Significant scientific statements and observations are without proper reference in the discussion. When explaining the outcomes, the author needs to be very cautious.

We have revised the the discussion section to avoid making any unsupported statements (page 17, lines 2-11; and page 23, lines 3-9). We have also cited proper references throughout the whole discussion to explain and support our findings.

- Author's needs to orient and provide the scientific justification to correlate all the parameters he performed, in a very scientific and significant manner.

Thanks for this useful comment; we have revised our discussion section accordingly. Please see page 17, lines 2-11, line 12, 15; page 18, lines 3-4; page 19, lines 1-3, and 8-11; page 20, lines 3-5, 18-19 and page 23, lines 4-8 .

- In both the discussion and the conclusion, the aims, rationale, and future perspectives are not evident clearly.

According to your useful comments, we have revised the discussion (page 17, lines 2-11) and conclusion (page 23, lines 2-8 ) to include study aims, rationale and future perspective.

- The discussion is usually organized at the beginning to address all the observations and evaluate them at the end. It makes the results easier to contextualize and simpler to comprehend. Furthermore, a minimal critical analysis should be provided.

We appreciate the reviewer’s comments and have revised the beginning of our discussion section to address all of the observation and provide a critical analysis. Please see page 17, lines 2-11. 

- According to the reviewer, a more thorough analysis of the findings is needed in the view of the current literature. To address the outcome measures/results separately and how they correlate with the existing literature, it would be better if the author restructured to take a more critical approach.

Thanks for the reviewer’s suggestion. We have revised our manuscript to address our results and how they correlate with the existing literature. We have restructured our discussion section. Please see page 17, lines 2-11, line 12, 15; page 18, lines 3-4; page 19, lines 1-3, and 8-11; page 20, lines 3-5, 18-19 and page 23, lines 4-8.

- The reviewer feels that the conclusion part is somewhere that seems out of focus. 

- The reviewer believes that the conclusion section appears to be out of sight somewhere.

We appreciate the reviewer’s comments and have reorganized our conclusion so that there is a greater focus on our findings. Please see page 23, lines 2-8.

---

## [Editor Report · Decision Letter 1]

16 Apr 2021

Predictors of Drug Survival for Biologic and Targeted Synthetic DMARDs in Rheumatoid Arthritis: Analysis from the TRA Clinical Electronic Registry

PONE-D-20-23397R1

Dear Dr. Chen,

We’re pleased to inform you that your manuscript has been judged scientifically suitable for publication and will be formally accepted for publication once it meets all outstanding technical requirements.

Kind regards,

Vineet Kumar Rai, PhD

Academic Editor

PLOS ONE
---

## [Editor Report · Acceptance letter]

22 Apr 2021

PONE-D-20-23397R1 

Predictors of Drug Survival for Biologic and Targeted Synthetic DMARDs in Rheumatoid Arthritis: Analysis from the TRA Clinical Electronic Registry 

Dear Dr. Chen:

I'm pleased to inform you that your manuscript has been deemed suitable for publication in PLOS ONE. Congratulations! Your manuscript is now with our production department. 

Kind regards, 

on behalf of

Dr. Vineet Kumar Rai 

Academic Editor

PLOS ONE